Clustering symptoms of non-severe malaria in semi-immune Amazonian patients

Martins Antonio C. 1
Araújo Felipe M. 1
Braga Cássio B. 1
Guimarães Maria G.S. 1
Nogueira Rudi 1
Arruda Rayanne A. 1
Fernandes Lícia N. 2
Correa Livia R. 2
Malafronte Rosely dos S. 2
Cruz Oswaldo G. 3
Codeço Cláudia T. 3
da Silva-Nunes Mônica 1 msnunes1@yahoo.com.br
1 Health Sciences Center, Federal University of Acre , Rio Branco, Acre , Brazil
2 Tropical Medicine Institute, University of São Paulo , São Paulo , Brazil
3 Scientific Computation Programm, Oswaldo Cruz Foundation , Rio de Janeiro, Rio de Janeiro , Brazil
Braga Erika
Electronic publication date: 2015 Oct 13
Publication date: 2015
Volume: 3
Electronic Location ID: e1325
Received 2015 Aug 4; Accepted 2015 Sep 24
Copyright: © 2015 Martins et al.
Copyright year: 2015
Copyright holder: Martins et al.
License: This is an open access article distributed under the terms of the Creative Commons Attribution License, which permits unrestricted use, distribution, reproduction and adaptation in any medium and for any purpose provided that it is properly attributed. For attribution, the original author(s), title, publication source (PeerJ) and either DOI or URL of the article must be cited.
License URL: https://creativecommons.org/licenses/by/4.0/

Keywords: Malaria, Symtoms, Amazon, Clustering

Funding: UFAC (Brazil) FUNTAC Edital FDCT 2011 grant # 14/2013 Edital PIBIC 01/2014 CNPq (Edital Universal 2011) FAPAC (Edital PPSUS 2013) 14/2013 UFAC Master’s Program This study received financial support from UFAC (Brazil), FUNTAC (Edital FDCT 2011), CNPq (Edital Universal 2011) and FAPAC (Edital PPSUS 2013) as well as support from the UFAC Master’s Program in Public Health. Research fellowships were awarded by CNPq, UFAC and FAPAC. The funders had no role in study design, data collection and analysis, decision to publish, or preparation of the manuscript.

==============================
Malaria is a disease that generates a broad spectrum of clinical features. The purpose of this study was to evaluate the clinical spectrum of malaria in semi-immune populations. Patients were recruited in Mâncio Lima, a city situated in the Brazilian Amazon region. The study included 171 malaria cases, which were diagnosed via the use of a thick blood smear and confirmed by molecular methods. A questionnaire addressing 19 common symptoms was administered to all patients. Multiple correspondence analysis and hierarchical cluster analysis were performed to identify clusters of symptoms, and logistic regression was used to identify factors associated with the occurrence of symptoms. The cluster analysis revealed five groups of symptoms: the first cluster, which included algic- and fever-related symptoms, occurred in up to 95.3% of the cases. The second cluster, which comprised gastric symptoms (nausea, abdominal pain, inappetence, and bitter mouth), occurred in frequencies that ranged between 35.1% and 42.7%, and at least one of these symptoms was observed in 71.9% of the subjects. All respiratory symptoms were clustered and occurred in 42.7% of the malaria cases, and diarrhea occurred in 9.9% of the cases. Symptoms constituting the fifth cluster were vomiting and pallor, with a 14.6% and 11.7% of prevalence, respectively. A higher parasitemia count (more than 300 parasites/mm3) was associated with the presence of fever, vomiting, dizziness, and weakness (P < 0.05). Arthralgia and myalgia were associated with patients over the age of 14 years (P < 0.001). Having experienced at least eight malaria episodes prior to the study was associated with a decreased risk of chills and fever and an increased risk of sore throat (P < 0.05). None of the symptoms showed an association with gender or with species of Plasmodium. The clinical spectrum of malaria in semi-immune individuals can have a broad range of symptoms, the frequency and intensity of which are associated with age, past exposure to malaria, and parasitemia. Understanding the full spectrum of nonsevere malaria is important in endemic areas to guide both passive and active case detection, for the diagnosis of malaria in travelers returning to non-endemic areas, and for the development of vaccines aimed to decrease symptom severity.

Introduction

Malaria features a broad spectrum of clinical outcomes that vary from asymptomatic infection (Da Silva-Nunes et al., 2012) to severe disease (O’Brien, Ramírez & Martínez, 2014), associated with profound anemia, jaundice, coma, plaquetopenia, respiratory distress, and cerebral malaria. Between asymptomatic presentation and cases that may progress to coma and death lies the clinical spectrum of nonsevere malaria, which generally occurs in semi-immune individuals.

The most easily recognizable symptoms in nonsevere malaria are chills, fever, and sweating, which are frequently observed in patients experiencing the first episode of malaria, a stage during which they have no immunity (López-Vélez et al., 1999). However, semi-immune patients may experience several other symptoms, such as headache, myalgia, arthralgia, weakness, abdominal pain and others, which can be helpful in the clinical diagnosis of nonsevere malaria (Da Silva-Nunes & Ferreira, 2007; Périssé & Strickland, 2008).

In general, diagnostic methods applied in malaria-endemic areas are heterogeneous in nature. In some countries, malaria is treated after a clinical diagnosis because of limited access to laboratory tests. To this effect, sensitivity and specificity of symptoms and signals are crucial for medical care because certain symptoms can be nonspecific and may be associated with other diseases; as a result, a patient may be misdiagnosed and receive the wrong treatment, which can have detrimental effects on his or her health (Ansumana et al., 2013; Cifuentes et al., 2013).

Other malaria-endemic areas use rapid tests, but such tests can yield false negative results, especially in the cases of low parasitemia counts, commonly seen in patients with asymptomatic infections (Okell et al., 2012). In most endemic settings, however, thick smears are widely used for malaria diagnosis because they are least expensive and easy to perform. However, even with well-trained human resources thick smears can yield false negative results in cases with low parasitemia.

Throughout Brazil, thick smears or rapid tests are administered at no cost to patients, and cases are detected using either passive or active methods. Passive methods are applied when a patient seeks diagnosis and treatment after experiencing a few days of sickness. Active methods are used by healthcare workers when examining potential cases among both sick and asymptomatic individuals. The Brazilian Ministry of Health (Ministério da Saúde, 2005) recommends that treatment be provided only after laboratory confirmation of malaria, either by thick smear, rapid diagnostic tests (RDTs), or, less commonly, by polymerase chain reaction (PCR).

Although the control of malaria requires the identification of a Plasmodium infection during a short prodromal asymptomatic period or during a long-lasting asymptomatic case (Matangila et al., 2014), testing every asymptomatic individual who lives in an endemic area and who could be in a prodromal period can be expensive; therefore, active case detection generally focuses on sick patients, especially in rural areas such as the Amazon, where one must travel a long distance to reach a patient’s house. Identifying the most common malaria symptoms, as well as the less frequent ones, can thus enhance the active detection strategy in settings with few resources and in which testing everybody is not feasible.

In addition to optimizing diagnosis and malaria control, the description of clinical heterogeneity in semi-immune individuals provides more accurate endpoints for vaccine efficacy studies, which can be used when evaluating semi-immune individuals in endemic countries (Roestenberg et al., 2012). Such vaccine trials may target the reduction of symptoms as a primary or secondary endpoint. For instance, an antimalarial vaccine may not prevent malaria, but the acquired immunity may reduce peripheral parasitemia and thus result in a less pronounced clinical picture. In this case, the quantification of symptoms and their reduction may be a valuable tool in determining vaccine efficacy.

Knowledge of all possible scenarios or clinical presentations of malaria, especially in the absence of fever, is important in the diagnosis of malaria in nonimmune and semi-immune individuals who live in malaria-free areas and who are seeking treatment. A review of malaria deaths occurring in several countries between 1985 and 2013 was performed, gathering one study published in 1985, 16 studies published between 1990 and 1999 and 34 studies published after 2000. This review revealed that 67.8% of the US travelers who died of malaria did not receive a correct diagnosis on the same day they presented at the inpatient service, and in 17.9% of the fatal cases, diagnoses were made only during the autopsy (Lüthi & Schlagenhauf, 2015). In the UK and in Switzerland, malaria-related deaths were more frequently observed within services whose medical staff members were not accustomed to seeing malaria patients (Lüthi & Schlagenhauf, 2015).

Materials and Methods

Study area and population

This study was conducted in Mâncio Lima, Acre, in the western Brazilian Amazon region. Mâncio Lima is 5,453 km2 in area and has 16,795 inhabitants living in urban (57.3%), rural or riparian (37.9%), and indigenous (4.8%) areas (IBGE, 2010). The city, located 38 km from Cruzeiro do Sul and 650 km northwest of Rio Branco, borders the municipality of Cruzeiro do Sul and Rodrigues Alves to the east, Amazonas state to the north, and Peru to the west. Mâncio Lima is an equatorial region surrounded by palm trees and rainforests (Acre, 2010). Its monsoon season occurs from November to April, with an annual rainfall of 1,600–2,750 mm. The city’s annual temperature ranges between 20 °C and 32 °C, and the annual relative humidity is 80–90% (Acre, 2010). In 2010, the human development index was 0.625. The economy’s main sources of income are cattle-raising, fishing, and producing and selling banana and cassava products.

In 2010, there were 5,729 autochthonous malaria cases within a total population of 16,795, resulting in an incidence rate of 34.11%. About 40% of those cases occurred in the urban areas of the city. In 2013, Mâncio Lima registered 6,936 cases of malaria, of which 29.1% were falciparum malaria and 70.3% vivax malaria; only 0.6% of the cases were of mixed species (Ministério da Saúde, 2015).

Population and sampling

Participants were sourced from malaria cases that were diagnosed using passive case detection at health posts in Mâncio Lima during the months of February, June, and November of 2012 and in May 2013. Except for June, all other months corresponded to the rainy season, when the cases are most prevalent. Patients with malaria symptoms and seeking a diagnosis were first submitted to a thick blood smear by health agents. Patients who tested positive and who were more than 5 years of age were asked to participate in the study; if they agreed to undergo an interview and blood sampling, they were included in the study. An age cutoff point of 5 years was chosen based on previous experience with questionnaire answers and graduating symptoms (Da Silva-Nunes & Ferreira, 2007). Only children who provided reliable information were included in the study. For minors, consent by the patient and by the patient’s parent or guardian was obtained. Participants were interviewed using a structured questionnaire that included demographic variables (e.g., gender and age), frequency of symptoms, and number of previous malaria episodes, among other factors. Blood samples were collected intravenously (when allowed) or by finger prick. All clinical interviews were performed by the same researcher (ACM), who had previously been trained by a malariologist (MdSN).

Blood sampling and laboratory diagnosis of malaria

Thick blood smears were stained with Giemsa. According to Ministry of Health guidelines (Ministério da Saúde, 2005), at least 100 fields were examined for malaria parasites by two experienced microscopists, using 700× magnification to conduct a semi-quantitative analysis of parasitemia (<200 parasites/mm3, 200–300 parasites/mm3, 301–500 parasites/mm3, 501–10.000 parasites/mm3, and >10.000 parasites/mm3). The slides were then reviewed by another experienced microscopist in an alternate city.

Venous blood was collected in sterile vacuum tubes with ethylenediaminetetraacetic acid (EDTA). Samples were centrifuged, and the sera were separated and stored at −20 °C until tested. In some cases, when the patient refused to have a venous blood sampling, digital blood from a finger prick was spotted onto FTA Micro Cards (Whatman, Clifton, New Jersey, USA) and was prepared as recommended by the manufacturer using FTA Purification Reagent (Whatman, Little Chalfont, UK). Blood samples were also examined using nested polymerase chain reaction (PCR)-based amplification of a species-specific segment of the 18S rRNA gene of human malaria parasites, as described by Kimura et al. (1997), with modifications by Win et al. (2002). The DNA templates for PCR amplification were isolated from 200 µl of whole blood (whenever available) using a Macherey–Nagel genomic DNA extraction kit for tissue (NucleoSpin Tissue; Macherey-Nagel, Düren, Germany).

Clinical assessment

Symptoms associated with malaria episodes were confirmed and assessed using microscopy and PCR, as described by Da Silva-Nunes & Ferreira (2007) but with certain modifications. Patients were asked to complete a structured questionnaire addressing the following common symptoms: fever, chills, sweating, headache, myalgia, arthralgia, abdominal pain, nausea, vomiting, dizziness, cough, diarrhea, weakness, inappetence, bitter mouth, pallor, coryza, sneezing, and sore throat. Although not a symptom, pallor was also included in the questionnaire, because this is a common complaint in malaria patients and is sometimes mistakenly perceived to be a signal of jaundice by inexperienced health personal, since in Portuguese the word “yellow” is commonly used by patients to refer to anemia or to anemic skin coloration. Patients were then asked if they noticed pallor at any time during their illness; if they responded positively, the pallor was called a “symptom.” The numerical scores of 0, 1, 2, and 3 were assigned to symptoms reported to be absent, mild, moderate, or severe, respectively. The author who conducted the interviews also assessed all infections to minimize inter-observer variation. Patients perceived each clinical manifestation (except fever) as absent, mild, moderate, or severe. Fever was classified as absent, mild, or severe. To minimize recall bias and the possibility that the drug treatment might interfere with their symptoms, patients were interviewed immediately after the diagnosis and before ingesting the first pills.

Treatment of malaria episodes

Patients were referred for treatment by authorized Ministry of Health workers, since, in Brazil, the drug is dispensed only by these healthcare workers. Patients with vivax malaria were treated with chloroquine and primaquine for 7 days, and those with falciparum malaria were treated with a combined regimen of mefloquine, artesunate, and primaquine or with a combination of artemether and lumefantrine (Coartem®) whenever possible, according to Brazil’s current malaria therapy guidelines (Ministério da Saúde, 2010).

Statistical analysis

Exploratory analysis

Data were entered using SPSS 13.0 software (SPSS Inc., Chicago, Illinois, USA). Parasite loads were stratified into two levels: (a) low parasitemia, ≤100 parasites in 100 fields (roughly corresponding to 1–300 parasites/µl of blood) and (b) moderate and high parasitemia, with >100 parasites in 100 fields (>300 parasites/µl of blood). Age was classified into <14 years and ≥14 years, in order to compare children and adolescents with adults. The number of previous confirmed malaria episodes was divided per ≤8 and >8 episodes, corresponding to the median number of previous episodes experienced by the subjects included in the study. The duration of symptoms in days was also divided per ≤3 and >3 days, corresponding to the median number of days during which symptoms were experienced. The distribution of the independent variables was identified using the student’s t-test to compare means and frequencies or proportions were compared using the chi-square test, with α = 0.05. The Somers’ D statistic measure (Somers, 1962) was used to test for a dose–response relationship between symptom scores and ordinal variables (level of parasitemia, age, duration of residence in Amazonia, number of recent malaria episodes, and duration of symptoms prior to interview).

Multiple correspondence analysis

Correspondence analysis is an exploratory and descriptive statistical technique used in analyzing data that are organized in contingency tables to verify associations or similarities between qualitative or quantitative variables; these variables are categorized without a probabilistic distribution defined a priori (Carvalho & Struchiner, 1992; Greenacre & Blasius, 2006). Multiple correspondence analyses facilitate the simultaneous examination of several variables; the results are graphically represented, so that each category of every variable is represented by a point, and the distance from one point to another represents the relationships among the variables’ categories (Carvalho & Struchiner, 1992; Mota, Vasconcelos & Assis, 2007).

The multiple correspondence analysis was initiated using an (n × p) matrix in which each (n) line corresponds to a patient and each (p) column refers to a studied characteristic. Each patient presents a (pi, i = 1, …, n) profile, defined by his/her characteristics; likewise, a (pj, j = 1, …, p) profile can be drawn for each variable on the basis of the patients’ answers (Greenacre & Blasius, 2006).

Considering the (n × p) matrix as a set of n points within a space of the p dimension, the center of gravity within the mass of data corresponds to the mean value of all profiles and therefore can be identified as the profile expected value. The distances between each point and the center of gravity represent those between observed and expected values, which are called χ2 distances (Greenacre & Blasius, 2006; Pereira, 2004).

The average of the χ2 distances corresponds to a measure of similarity called inertia; it takes the value of zero when all data matrix points are superimposed on the center of gravity. The total inertia is decomposed as relative inertias pertaining to each evaluated dimension (Pereira, 2004; Paula et al., 2010).

The square root of the inertia corresponds to an eigenvalue, which indicates the total amount of variability in the data that is explained by that dimension (Pereira, 2004).

The analysis of the absolute contribution of each category obtained through the inertia and the observation of the graph points for the correspondence analysis allow for the conceptual characterization of a graph’s axis, also known as dimension. In turn, the relative contribution of a category measures how much of the variability of a given category is explained by the analyzed dimension (Mota, Vasconcelos & Assis, 2007).

In the present study, it was expected that the graphical representation of the two dimensions would illustrate the grouping areas of the categories of variables, which were included in the analysis on the frequency of malaria-related symptoms. This would allow the identification of symptoms that are grouped according to specific characteristics (age, gender, Plasmodium species, number of previous malaria episodes, and parasitemia).

To complement the interpretation of the results yielded by the multiple correspondence analysis, a dendrogram was designed to divide the data into similar groups on the basis of the coordinates’ average (Maechler et al., 2015).

The statistical procedures were performed using the free R software environment, version 3.1.1 (The R Foundation for Statistical Computing, Vienna, Austria; http://www.r-project.org/), using the FactoMineR package (Husson et al., 2014).

Univariate logistic regression

Univariate logistic regression analysis using R software version 3.1.1 was implemented to quantify the probability of experiencing each symptom on the basis of Plasmodium species, parasitemia, age, number of previous malaria episodes, and number of days during which symptoms were experienced.

Ethical considerations

This study was approved by the Ethics Committee for Research with Human Beings at the Federal University of Acre (protocol number 23107.016975/2011-28). Written informed consent was obtained from each participant or by his/her parent prior to the study.

Results

Clinical, parasitological, and epidemiological characteristics of malaria cases

A total of 173 malaria cases were selected on the basis of a positive thick smear, indicating either P. vivax or P. falciparum. Samples from all patients were submitted for a molecular diagnosis, of which 171 cases revealed single-species malaria (137 cases of P. vivax and 34 cases of P. falciparum). Two cases, diagnosed as P. vivax based on a thick smear, reported a molecular diagnosis of mixed infection (P. falciparum and P. vivax) and were therefore excluded from the analysis.

Of the selected cases, 55.6% were male, and 44.4% were female; the participants’ ages ranged between 6 and 93 years (median = 26 years, average = 29.3 years). The majority of the people were born in the Amazon, and the average duration of residence in the area was 28.4 years. Only 3.03% of the participants did not have malaria prior to the study, and, for 7.88%, the malaria episode experienced during the present study was their second occurrence. Almost 50% of the patients reported having at least eight malaria episodes prior to the study. Forty percent of the cases had patent, but low, parasitemia (lower than 200 parasites/mm3); 39.40% had parasitemia between 200 and 500 parasites/mm3; and only two cases (1.20%) had parasitemia higher than 10.000 parasites/mm3 (Table 1). The median duration of symptoms was 3 days, while the mean was 5.33 days. The median duration of symptoms for each parasitemia strata was also 3 days.

Table 1 Epidemiological and parasitological characteristics of malaria cases, Mâncio Lima, 2012–2013.

Variables	n	%	
Gender			
Female	76	44.4	
Male	95	55.6	
Type of malaria			
P. vivax	137	80.10	
P. falciparum	34	19.10	
Parasitemia			
<200 parasites/mm3	68	40.00	
200–300 parasites/mm3	44	25.90	
301–500 parasites/mm3	23	13.50	
501–10.000 parasites/mm3	33	19.40	
10.001–100.000 parasites/mm3	2	1.20	
Number of previous malaria episodes			
None	5	3.03	
Only one malaria episode	13	7.88	
Between 2 and 7 malaria episodes	66	40.00	
More or equal to 8 malaria episodes	81	49.09	

Table 2 shows the frequency of each symptom, and Fig. 1 shows the intensity of these symptoms. The most frequently experienced symptoms were headache (86.5%), fever (78.4%), and chills (75.4%), while the least frequent was diarrhea (9.9%).

Table 2 Frequency of 19 symptoms related to malaria, Mâncio Lima, 2012–2013.

Group of symptoms	Symptom	P.v n (%)	P.f n (%)	P.v + P.f n (%)	
Algic and fever-related symptoms	Headache	119 (86.9)	29 (85.3)	148 (86.5)	
Fever	110 (80.3)	24 (70.6)	134 (78.4)	
Chills	104 (75.9)	25 (73.5)	129 (75.4)	
Myalgia	86 (62.8)	24 (70.6)	110 (64.3)	
Arthralgia	85 (62.0)	24 (70.6)	109 (63.7)	
Weakness	88 (64.2)	19 (55.9)	107 (62.6)	
Sweating	75 (54.7)	19 (55.9)	94 (55.0)	
Dizziness	67 (48.9)	22 (64.7)	89 (52.0)	
Any symptom	131 (95.6)	32 (94.1)	163 (95.3)	
Gastric symptoms	Nausea	57 (41.6)	16 (47.1)	73 (42.7)	
Bitter mouth	48 (35.0)	16 (47.1)	64 (37.4)	
Inapetence	48 (35.0)	15(44.1)	63 (36.8)	
Abdominal pain	46 (33.6)	14 (41.2)	60 (35.1)	
Any symptom	94 (68.5)	29 (85.3)	123 (71.9)	
Respiratory symptoms	Cough	32 (23.4)	9 (26.5)	41 (24)	
Coryza	33 (24.1)	5 (14.7)	38 (22.2)	
Sore throat	33 (24.1)	3 (8.8)	36 (21.1)	
Sneeze	26 (19.0)	5 (14.7)	31 (18.1)	
Any symptom	61 (44.5)	12 (35.3)	73 (42.7)	
Vomiting and pallor	Vomiting	19 (13.9)	6 (17.6)	25 (14.6)	
Pallor	14 (10.2)	6 (17.6)	20 (11.7)	
Any symptom	29 (21.2)	8 (23.5)	37 (21.6)	
Diarrhea	Diarrhea	11 (8.0)	6 (17.6)	17 (9.9)	

Figure 1 Frequency and intensity of each malaria-related symptom in Mâncio Lima (2012–2013).

Numbers on y axis are percentages. The shading pattern of each bar indicates the proportion of patients reporting a given symptom as absent, mild, moderate or severe. C1, Algic and fever-related symptoms; C2, Gastric symptoms; C3, Respiratory symptoms; C4, Vomiting and pallor; C5, Diarrhoea.

Multiple correspondence analysis and identification of symptom clusters

Figures 2A and 2B show the joint distribution of the correspondence analysis dimensions; three qualitative variables are included in the analysis: parasitemia, number of previous malaria episodes, and malaria species. The first two dimensions of the multiple correspondence analysis explained 20.3% and 11.4% of the total data variability, respectively. Symptoms are represented as “yes” and “no,” indicating the influence of the presence of each symptom and the influence of the absence of each symptom. As depicted in Table 3 and Figs. 2A and 2B, it is possible to verify whether the following variable categories have an absolute contribution higher than 10% over dimension 1: fever, chills, sweating, headache, nausea, vomiting, abdominal pain, arthralgia, dizziness, myalgia, inappetence, weakness, and bitter mouth. The presence of these symptoms is graphically represented in the left lower quadrant, along the x axis. In dimension 2, the categories that stood out were respiratory symptoms: cough, coryza, sneezing, sore throat, and diarrhea. They are graphically represented in the upper portion of the left upper quadrant, along the y axis.

Figure 2 Joint distribution of the correspondence analysis dimensions for qualitative variables (parasitemia, number of previous malaria episodes and malaria species).

(A) Joint distribution of the correspondence analysis dimensions for qualitative variables (parasitemia, number of previous malaria episodes and malaria species). Dimension 1 and 2 contribute to explain 31.74% of the variance. The contribution of each symptom in dimension 1 and 2 is represented as a triangle, where ‘yes’ indicates the contribution of the presence of the symptom, and ‘no’ indicates the contribution of the absence of the symptom. Subjects are represented as blue dots, and their position in each quadrant demonstrate the referred symptom profile of each individual (near the presence of symptoms or near the absence of symptoms). (B) Joint distribution of the correspondence analysis dimensions for qualitative variables (parasitemia, number of previous malaria episodes and malaria species) with circles indicating groups of symptoms identified in the cluster analysis. C1, Algic and fever-related symptoms; C2, Gastric symptoms; C3, Respiratory symptoms; C4, Vomiting and pallor; C5, Diarrhoea.

Table 3 Relative contributions of the first and second dimensions of the correspondence analysis and R2 values according to symptoms, Mâncio Lima, 2012–2013.

Group of symptoms	Symptom	Dimension 1	Dimension 2	R2 dimension 1	P value 1	R2 dimension 2	P value 2	
Algic and fever-related symptoms	Headache	0.279	0.016	0.279	<0.001			
Fever	0.409	0.005	0.409	<0.001			
Chills	0.370	0.026	0.370	<0.001	0.026	0.04	
Myalgia	0.273	0.003	0.273	<0.001			
Arthralgia	0.274	0.052	0.274	<0.001	0.052	0.003	
Weakness	0.305	0.007	0.305	<0.001			
Sweating	0.160	0.002	0.160	<0.001			
Dizziness	0.329	0.035	0.329	<0.001	0.035	0.016	
Gastric symptoms	Nausea	0.284	0.001	0.284	<0.001			
Bitter mouth	0.317	0.001	0.317	<0.001			
Inapetence	0.291	0.001	0.291	<0.001			
Abdominal pain	0.153	0.027	0.153	<0.001	0.027	0.035	
Respiratory symptoms	Cough	0.046	0.443	0.046	0.006	0.443	<0.001	
Coryza	0.003	0.495		<0.001	0.443	<0.001	
Sore throat	0.001	0.461		<0.001	0.461	<0.001	
Sneeze	0.009	0.511			0.511	<0.001	
Vomiting and pallor	Vomiting	0.161	0.006		0.161			
Pallor	0.123	0.001		0.123			
Diarrhea	Diarrhea	0.033	0.100		0.033	0.100	<0.001	

The visual interpretation of the combined representation of dimensions 1 and 2 in Figs. 2A and 2B and the output in Table 3 indicate that the subjects represented as blue dots in the left lower quadrant present most of the symptoms investigated; patients represented in the right lower quadrant do not show most of the symptoms investigated, and patients represented in the upper two quadrants tend to refer respiratory symptoms more often.

The dendrogram (Fig. 3) classifies the symptoms displayed in Fig. 2A in five clusters, which are also represented in Fig. 2B. The first cluster of symptoms (C1) includes fever-related symptoms and most of the algid symptoms (headache, chills, sweating, arthralgia, myalgia, weakness, and dizziness). The most frequently experienced symptoms were headache (86.5%), fever (78.4%), and chills (75.4%). Arthalgia (63.7%), myalgia (64.3%), and weakness (62.6%) were also frequent. Symptoms of this group occurred in up to 95.3% of the cases. The second cluster of related symptoms (C2) comprised gastric symptoms (nausea, abdominal pain, inappetence, and bitter mouth). They occurred in frequencies of 35.1–42.7%, and 71.9% participants had at least one of these symptoms. Respiratory symptoms (from both the upper and lower respiratory tracts) were clustered (C3). Coryza, sneezing, and sore throat were reported in 18.1–22.2% of the cases, and cough was present in 24.0% of the patients. Symptoms of this group occurred in up to 42.7% of the malaria cases. Diarrhea (C5) did not cluster with any other symptom, and it occurred in 9.9% of the cases. Vomiting and pallor constituted the fifth cluster of symptoms/signs (C4), with prevalences of 14.6% and 11.7%, respectively. Of all the symptoms, diarrhea, vomiting, and pallor were the three that occurred least frequently.

Figure 3 Dendrogram of the coordinates of the first two dimensions of the correspondence analysis with clusters of groups.

C1, Algic and fever-related symptoms; C2, Gastric symptoms; C3, Respiratory symptoms; C4, Vomiting and pallor; C5, Diarrhoea.

Figure 4 shows the correspondence analysis result classified by the Plasmodium species, parasitemia, and the number of previous malaria episodes. The confidence ellipse delimitates the centroid around each variable. A visual examination of the joint distribution of the first two dimensions revealed no difference in the median profile of vivax and falciparum malaria; that is, the ellipses are superimposed. However, the symptom profile varies by the number of parasitemia and that of previous malaria episodes. Patients with a higher parasitemia count (≥300 parasites/mm3) have more symptoms (represented in the left side of dimension 1), and those with a lower parasitemia count (<300 parasites/mm3) tend to have fewer symptoms (represented on the right-hand side of dimension 1). Finally, patients with more than eight previous malaria episodes generally experienced fewer symptoms, except for sore throat, which was more pronounced in these patients.

Figure 4 Multiple correspondence analysis classified by Plasmodium species, parasitemia and number of previous malaria episodes.

The confidence ellipse delimitates the centroid around each variable. For Parasitemia, ellipses are separated, indicating a good distinction in symptom profile between low and high parasitemia. Patients with higher parasitemia (≥300 parasites/mm3) have more symptoms (pink ellipse is seen in the left-hand side of dimension 1) and those with lower parasitemia (<300 parasites/mm3) tend to have fewer symptoms (blue ellipse represented on the right-hand side of dimension 1). For Plasmodium species, ellipses are superimposed, indicating that the symptom profile shown in each type of malaria is very similar, and therefore malaria species cannot be differentiated using symptoms only. For number of previous episodes of malaria, there is little superimposition of the ellipses, indicating that it is possible to separate between those patients with more than eight previous malaria episodes, who generally experience fewer symptoms (pink ellipse is in the right side of Dimension 1, indicating less symptoms), except for symptoms sore throat, which is more pronounced in these patients (note that the pink ellipse is above the blue ellipse in Dimension 2, indicating that symptoms distributed along dimension 2 are differentiating patients with 8 or more episodes from those with less than 8 episodes).

Factors associated with the clinical and parasitological features of malaria

Using the Wald test for logistic regression (Table 4 and Fig. 5), a higher parasitemia count (>300 parasites/mm3) was associated with the presence of fever (OR = 5.4, 95% CI [1.81–16.12], P = 0.003), vomiting (OR = 2.51, 95% CI [1.06–5.94], P = 0.036), dizziness (OR = 2.21, 95% CI [1.14–4.26], P = 0.018), and weakness (OR = 2.74, 95% CI [1.33–5.63], P = 0.006). A higher parasitemia count, however, was not associated with other symptoms. The occurrence of arthralgia (OR = 5.44, 95% CI [2.37–12.52], P < 0.001) and myalgia (OR = 5.64, 95% CI [2.45–12.99], P < 0.001) was associated with age, being more likely to occur in patients older than 14 years. Having experienced at least eight malaria episodes prior to the study was associated with a decreased risk of fever (OR = 0.30, 95% CI [0.18–0.84], P = 0.017) and chills (OR = 0.44, 95% CI [0.21–0.91], P = 0.027) and with an increased risk of sore throat (OR = 2.36, 95% CI [1.1–5.04], P = 0.027). None of the symptoms showed an association with gender, Plasmodium species, or duration of symptoms.

Figure 5 Relationship between symptom intensity and parasitological features (number of previous malaria episodes, parasitemia and age) in Mâncio Lima (2012–2013).

Numbers on y axis are percentages. The shading pattern of each bar segment indicates the proportion of patients reporting a given symptom as absent, mild, moderate or severe. P value = Somers’ d test for ordinal variables.

Table 4 Association between symptoms and clinical features in malaria cases, Mâncio Lima, 2012–2013.

	OR	95% CI	P value	
Outcome: parasitemia higher than 300 parasites/mm 3	
Presence of fever	5.4	1.81–16.12	0.003	
Presence of vomiting	2.51	1.06–5.94	0.036	
Presence of dizziness	2.21	1.140-4.26	0.018	
Presence of weakness	2.74	1.33–5.63	0.006	
Outcome: age older than 14 years old	
Presence of arthralgia	5.44	2.37–12.52	<0.001	
Presence of myalgia	5.64	2.45–12.99	<0.001	
Outcome: number of previous malaria equal or higher than 8	
Presence of fever	0.3	0.18–0.84	0.017	
Presence of chills	0.44	0.21–0.91	0.027	
Presence of sore throat	2.36	1.1–5.04	0.027	

Severe headache, myalgia, chills, and fever were observed in more than one third of the cases, but all the other symptoms were mild or moderate (Fig. 1). There was no association between symptom intensity and gender or malaria species.

The intensity of chills, fever, and sweating was associated with the number of previous episodes of malaria, and patients who had experienced more than eight episodes of malaria had less intense symptoms (P < 0.005). A higher parasitemia count was associated with more intense fever (P < 0.001), while myalgia was more intense in patients over the age of 14 years (P < 0.001) (Fig. 5).

When analyzing the intensity of symptoms with the duration of symptoms, there were associations with chills, fever, sweating (P = 0.002), headache (P < 0.001), nausea (P < 0.001), abdominal pain (P < 0.001), diarrhea (P < 0.008), and arthralgia (P = 0.032), which tended to be more intense in patients with shorter durations of illness than in those experiencing the disease for more than 3 days.

Discussion

Malaria is a clinical condition that generates several symptoms. This study performed a hierarchical analysis that grouped these symptoms into five clusters. Two of the clusters combine the most common symptoms observed in nonsevere malaria (the cluster of algic and fever-related symptoms and the cluster of gastric-related symptoms). These symptoms have been widely reported in many clinical and experimental studies. The remaining three clusters focus on respiratory symptoms and diarrhea, vomiting, and pallor.

In this study, headache was the most frequent symptom, a finding that has been reported in previous studies of African children (Roestenberg et al., 2012) and pregnant women (Tahita et al., 2013) with malaria. Fever and related symptoms (chills and sweating) were also frequently observed. However, fever was less likely to occur in patients with a history of multiple malaria episodes and a low parasitemia count.

The analysis of symptom intensity showed that a higher parasitemia count is associated with higher fever, and a lower parasitemia count is associated with mild fever or, sometimes, an absence of fever. Studies have reported a relationship between parasitemia and symptom intensity (Da Silva-Nunes & Ferreira, 2007; Torres et al., 2014). The lack of fever can hinder healthcare delivery because, more often than not, an active malaria diagnosis is fever-oriented (Cifuentes et al., 2013). Also, symptoms tended to be more intense in the first days of illness, probably because of the semi-immune status of the patients who had previously experienced episodes of malaria. Thus, healthcare workers in endemic areas must receive formal training on the clinical spectrum of malaria, including nonsevere malaria without fever, and on the heterogeneity of symptoms during illness.

Arthalgia and myalgia were common and were more likely to occur in patients over the age of 14 years. Other studies (Da Silva-Nunes & Ferreira, 2007; Chandramohan et al., 2001) have confirmed this finding and have revealed a higher susceptibility to skeletal pain in older individuals.

Respiratory symptoms associated with nonsevere malaria have been reported for vivax malaria. In a study of 15 adults with uncomplicated vivax malaria, a total of 40% reported limited dry cough (Anstey et al., 2002). Flu-like symptoms such as coryza and sneezing have also been observed. However, it is difficult to distinguish between malaria flu-like symptoms and influenza. Experimental malaria studies have shown that flu-like symptoms in malaria can be easily classified as flu itself (Lillie et al., 2012). Thus, it must be underscored that flu-like symptoms can be observed in cases of malaria, and in areas where malaria transmission is present, testing for malaria is important in patients presenting with flu-like symptoms at health-centres.

Gastrointestinal symptoms associated with malaria have been reported in several studies (He et al., 2013; Luxemburger et al., 1998). While some have found gastric symptoms to be effective predictors of malaria, especially in falciparum malaria (He et al., 2013), others have found them to be infrequent (Luxemburger et al., 1998). However, these studies tend to group nausea, diarrhea, and vomiting. The hierarchical clustering analysis performed in the present study showed that while the majority of gastric symptoms are experienced at the same time, diarrhea and vomiting form a separate cluster of symptoms and are not related to nausea and other minor gastric symptoms. This is a novel finding, but the reasons for that are unclear.

This study clusters vomiting with pallor, which confirms the finding reported in a study of African children with malaria (Vinnemeier et al., 2012). In some studies, vomiting was associated with falciparum malaria, but this can be a result of the higher parasitemia count caused by P. falciparum, because these cases had a much higher parasite load (Luxemburger et al., 1998). Although no relationship between vomiting and the Plasmodium species was found, an association between vomiting and a higher parasitemia count was observed.

Diarrhea was infrequent in semi-immune patients and was more closely related to vomiting and pallor than to any other symptom. In nonimmune patients, an increased prevalence of diarrhea may occur. Taylor et al. (2010) found a 21% prevalence in the meta analyses of malaria cases of travelers returning from endemic areas for malaria. For military personnel working in endemic areas, diarrhea was reported to be at a 50% prevalence (He et al., 2013). Even in Brazilian travelers returning from the Amazon, diarrhea was reported at higher frequencies (67% for falciparum and 55% for vivax cases) (Dos-Santos et al., 2014). Thus, diarrhea can be related to parasitemia and/or to acquired immunity. In areas where other febrile illnesses co-exist, diarrhea can be frequent but have low specificity for malaria (Vinnemeier et al., 2012).

Nausea, inappetence, bitter mouth, and abdominal pain were relatively frequent (up to 68.5%); however, few studies have cited them as frequent symptoms. Mutanda et al. (2014) reported inappetence in children as an effective predictor of malaria. An examination of these discrete gastric symptoms can help to assess malaria in specific groups, such as children, or to provide a more detailed evaluation of vaccine efficacy. Bitter mouth and nausea can be an organism’s natural defense against toxic substances (Peyrot des Gachons et al., 2011) and response to malarial toxins (Haldar & Mohandas, 2009; Hiller et al., 2004)).

In the present study, both the frequency and intensity of certain symptoms varied by past exposure to malaria. The higher the number of previous malaria episodes, the less intense the presentation of certain symptoms, such as fever, chills, and sweating. The data suggest that previous episodes confer some amount of clinical immunity, and symptoms can therefore become less pronounced.

The quantification of symptoms in nonsevere malaria is of increasing importance, given that several vaccines are being tested on human subjects. These vaccines may reduce the intensity of symptoms, and thus the optimization of an algorithm to assess symptoms can provide better estimates of vaccine efficacy, especially in the case of developing a vaccine for P. vivax (Arévalo-Herrera et al., 2014).

Nevertheless, this study has certain limitations. We were unable to identify symptoms that could differentiate noncomplicated vivax and falciparum malaria, although a limited number of P. falciparum cases were assessed. However, Luxemburger et al. (1998) conducted a study in Thailand using a larger dataset and did not find differences in symptoms between the two species. McKenzie et al. (2006) revealed that falciparum malaria showed higher incidence rate of fever than vivax malaria; however, this was related to the parasitemia levels. In areas where P. vivax and P. falciparum co-exist, the clinical pictures may overlap, making it impossible to differentiate between the two parasites without diagnostic tests.

Conclusion

Malaria is a complex disease that presents with multiple symptoms. The nonsevere form features a broad range of symptoms that can be grouped into clusters, with some symptoms appearing more frequently than others. The existence and intensity of symptoms are related to the patient’s age, as well as to parasitemia level and exposure to the Plasmodium species. This information is important for healthcare workers in endemic areas who perform both passive and active diagnoses. In addition, a complete assessment of malaria-related symptoms is useful for the diagnosis of malaria in travelers returning to nonendemic areas. The evaluation of effects of the parasite-host relationship on clinical presentation can be useful for studies assessing immunological response and the development of vaccines aimed at decreasing symptom severity. New statistical techniques, such as multiple correspondence analysis, as shown in this study, can help to identify groups of patients with similar clinical presentation according to variables of interest, in malaria studies as well as in studies of other diseases.

Supplemental Information

Supplemental Information 1 Raw data

Click here for additional data file.

We thank the population and the local health and government authorities for their help.

Abbreviations

Hea headache

Fev fever

Chi chills

Mya myalgia

Art arthralgia

Wea weakness

Swe sweating

Diz dizziness

Nau nausea

Bit bitter mouth

Ina inappetence

Abd abdominal pain

Cou cough

Cor Coryza

Sor sore throat

Sne sneezing

Pal pallor

Dia diarrhoea

Additional Information and Declarations

Competing Interests

Author Contributions

Human Ethics

The authors declare there are no competing interests.

Antonio C. Martins conceived and designed the experiments, performed the experiments, analyzed the data, wrote the paper, prepared figures and/or tables, reviewed drafts of the paper.

Felipe M. Araújo, Cássio B. Braga, Maria G.S. Guimarães, Rudi Nogueira, Rayanne A. Arruda, Lícia N. Fernandes and Livia R. Correa performed the experiments, reviewed drafts of the paper.

Rosely dos S. Malafronte performed the experiments, contributed reagents/materials/analysis tools, reviewed drafts of the paper.

Oswaldo G. Cruz analyzed the data, wrote the paper, reviewed drafts of the paper.

Cláudia T. Codeço analyzed the data, wrote the paper, prepared figures and/or tables, reviewed drafts of the paper.

Mônica da Silva-Nunes conceived and designed the experiments, analyzed the data, wrote the paper, reviewed drafts of the paper.

The following information was supplied relating to ethical approvals (i.e., approving body and any reference numbers):

Ethics Committee for Research with Human Beings at the Federal University of Acre, approval number 23107.000960/2010-67.

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
