# Peer review of "Clustering symptoms of non-severe malaria in semi-immune Amazonian patients"

_PeerJ, doi:10.7717/peerj.1325_

## Round 0.1 · original submission · Minor Revisions

The manuscript is well written and well defined which presents clear aims. Multiple correspondence analysis and hierarchical cluster used to identify clusters of symptoms among semi-immune individuals exposed to malaria in the Brazilian Amazon endemic region is a novelty which could be an important reference in the literature.

Reviewer 1 ·

Basic reporting

The study is of low relevance and just interested people who work and research on the basic theme of the study. The manuscript is well presented and written. Tables and figures are correctly presented and are relevant to the understanding of the study. Written English is acceptable. References are sufficient and were well cited.

Experimental design

The study has clear objective, it was well designed and well analyzed. The discussion is closely related to the goals and results of the study. The study methodology is appropriate. Statistical analysis is original to the theme of the study and is very well presented. The research has been conducted in conformity with the prevailing ethical standards in the field

Validity of the findings

The discussion was consistent and based solely on the results of statistical analysis.The main conclusion of the study was well defined and speculative arguments are appropriate.

Additional comments

Martins CA and colleagues present a new way to analyze symptoms reported by semi-immune patients with uncomplicated malaria. There is nothing new in its results, except the statistical methods used to do so. Has long been known that symptoms reported by patients are not relevant for the diagnosis of infection in the patient. However, considering a future malaria vaccine that targets the reduction of symptoms, the results of this study may be useful in the evaluation of its efficacy.

·

Basic reporting

This is an interesting analysis that attempts to cluster common symptoms of malaria into defined groups and assess the frequency with which they occur, and stratify by age, infection parasite density and history of infection.

This study reveals the importance of often-neglected symptoms of malaria and the use of multiple correspondence analysis for clustering of symptoms is novel and useful for identifying groups of infected individuals with similar clinical presentations, both for individuals in endemic areas and travellers returning to non-endemic countries.

A few minor suggestions are provided for the authors' consideration:

Introduction:
Lines 73-76 (as per the PeerJ-generated manuscript): The authors state that rapid tests for malaria often yield false negative results in patients with low parasitemia counts, but counter with, “in most endemic settings, however, thick smears are widely used for malaria diagnosis”. This statement is somewhat misleading and suggests that thick smears are immune to yielding false negative results, which is fundamentally untrue.
Lines 101-109: It would be useful to know the time period and scale of data in the Lüthi & Schlagenhauf review.
Lines 117-122: References for the climate information for Mancio Lima would be useful.
Lines 376-378: It would be useful to additionally stress the importance of diagnostic testing for patients presenting with flu-like symptoms at health-centres.


Discussion:
Lines 387-388: The Vinnemeier et al., 2012 reference needs to be moved to the end of the sentence.

Figures:
Line 692: The caption for figure 2 is incomplete (cuts off mid-sentence).

Experimental design

Methods:
Lines 189-190: What is the motivation for stratification into more and fewer than 100 parasites/100 fields, and for stratification of age? Are the cut-off points arbitrary or informed? If informed, by what?

Validity of the findings

Discussion:
Lines 425-427: The statement, “In areas where P. vivax and P. falciparum co-exist, the clinical pictures may overlap, making it impossible to differentiate between the two parasites except when the parasitemia levels differ”, is confusing. While it is true that the spectrum of symptoms has a lot of overlap between the two species of Plasmodium, the parasite density (parasitemia levels) in each infection is not relevant for differentiating between species for two reasons: i) while it is true that P. falciparum infection typically results in higher parasite densities, this is not always the case – particularly in semi-immune individuals, and ii) parasite density would never be the only method used for parasite-based diagnosis of malaria, as the two species of malaria are differentiable both under the microscope and by using rapid diagnostic tests.

---

## Round 0.2 · accepted · Accept

The manuscript was properly reviewed addressing all of the points raised by the Reviewer#2.